# Do environmental, social, and governance scores improve green innovation? Empirical evidence from Chinese-listed companies

Chunlian Zhang[1,2‡], Danni Chen[3‡]*

1 School of Economics and Trade, Nanchang Institute of Technology, Nanchang, Jiangxi, China, 2 The Water Economy and Water Rights Research Center, Nanchang Institute of Technology, Nanchang, Jiangxi, China, 3 School of Finance, Jiangxi University of Finance and Economics, Nanchang, China

‡ DC and CZ have contributed equally to this work and share first authorship.
* 2202121925@stu.jxufe.edu.cn

**Data Availability Statement:** All relevant data are within the manuscript and its Supporting Information files. All files are available from

## Abstract

Environmental, social, and governance (ESG) has become a buzzword in investment circles as ecological damage and climate warming occur. ESG assessment is one of the important institutions of the green financial system, which plays a significant part in boosting corporate green development. We use the number of green patent applications and green patent citations to measure corporate green innovation and analyze the micro-green effects of the ESG score system using the panel fixed effects models, which means that we explore the impact of the ESG scores on corporate green innovation performance, the specific mechanism of this effect, and the asymmetry of this impact under different moderation effects by using Chinese listed A-shares in Shanghai and Shenzhen from 2010–2019 as our research sample. We find that ESG positively affects corporate green innovation; the higher the ESG evaluation, the more it improves firms' green innovation performance. The promotion effect is reflected quantitatively and qualitatively and remains valid after several robustness tests. In addition, the contribution of ESG to corporate green innovation is achieved through two main paths improving corporate investment efficiency and government-enterprise relations. Corporate black attributes inhibit the contribution of ESG to green innovation, while green attributes strengthen the contribution of ESG to green innovation performance. Our study demonstrates the importance of corporate participation in environmental, social, and governance practices for corporate green innovation, which is beneficial for achieving win-win environmental, social, and economic results. Furthermore, our research completes the research on the effects of corporate green performance and green finance. It can provide empirical references for promoting corporate green development and improving the ESG evaluation system.

## Introduction

Green innovation mainly emphasizes the sustainability of innovation, which describes novel products, processes, and techniques that might minimize the dangers to the environment,

thefigshare database (10.6084/m9.figshare.
22578787 10.6084/m9.figshare.22578808).

**Funding:** The authors acknowledge the financial support from the project of the Water Economy and Water Rights Research Center, a school-level platform in Nanchang Institute of Technology: An empirical study on the Microeconomics of ESG performance under the 'Dual-carbon' vision (22ZXZD01). The funders had no role in study design, data collection and analysis, decision to publish, or preparation of the manuscript.

**Competing interests:** The authors have declared that no competing interests exist.

pollution, and resource consumption throughout their life cycles [1–3]. Moreover, green innovation is an essential form for companies to practice the environmental, social, and governance concepts and an important tool to drive the green transformation of enterprises [4]. Green innovations for companies are low-carbon, energy efficient, and effective [5], but it also has characteristics such as long-term riskiness, public goods, and positive environmental externalities [6]. With the increasing economic globalization and industrialization, the natural world is subject to significant adverse impacts. Environmental pollution problems are becoming more prominent, severe climate problems are becoming more powerful, and green innovation may be essential to reconcile the contradictions between man and nature [7, 8].

Several elements drive corporate green innovation performance, as businesses need to maintain a competitive edge and increase corporate value. The literature has classified the factors influencing green innovation into four categories. The first is market factors, including market pressure, green consumer demand, capital market opening, and environmental labeling certification [9–14]. Corporate green innovation will be aided by the news media and public social supervision [15]. The second is environmental policy issues. Some studies have indicated that these rules can encourage corporate green innovation [16]. Some studies have discovered a link between environmental restrictions that first inhibit corporate green innovation and later promote it [17, 18]. Others have discovered pilot policies of emissions trading [19], low-carbon pilot policies [20, 21], emission permit systems [22], carbon emission trading systems [23, 24], green credit policies [25–27], clean production audit (CPA) program [28] and environmental information disclosure system [29–32] can stimulate enterprises to make green innovation. The third is the political-enterprise relationship, which manifests as political affiliation and government subsidies. Political affiliation inhibits firms' green innovation, especially when the market degree is low [33]. And subsidies have a driving influence on corporate green innovation performance. However, political affiliation encourages enterprises' green innovation by raising R&D spending and organizational capital [34, 35], but some studies show no significant relationship between corporate subsidies and green innovation [36]. The fourth is internal corporate factors, CEO responsible leadership [3], executive academic experience [37], sustainability goals [38], CSR performance [4], and internal controls of institutional investors all contribute to encouraging green business innovation.

In a broad sense, environmental, social, and governance (ESG) can be seen as an extension of corporate social responsibility because it uses the three criteria of environmental, social, and internal governance to evaluate businesses [39], which reflects the degree of green transformation, and environmental image of enterprises [40]. ESG is also an ESG investment concept pursued by investors and becomes an investment basket of ESG factors. Hence, the ESG concept gradually becomes the consensus of global enterprises, investors, and financial institutions [41, 42]. ESG is an essential indicator of corporate green development, which is gradually incorporated into corporate development strategies [43, 44]. Current research has focused on the link between ESG and company performance. Some argue that ESG is unrelated to corporate profitability, cost of capital, or ESG deteriorates corporate performance [45]. Other researchers find that ESG scores can alleviate firms' financing constraints and improve their business performance [46–49]. In addition, better business stock returns with correspondingly higher stock liquidity and a dampening effect on crash risk are linked to higher ESG ratings [50, 51]. ESG ratings can help firms improve innovation performance and corporate value [52, 53]. However, other research believes ESG has a detrimental effect on corporate value [54]. Furthermore, a study found ESG investors in the Asia-Pacific region and the US perform similarly to the market. ESG investments are more suitable for 'value-driven investors' (VDI). It also found that European investors will pay the price for making ESG investments, which is not conducive to improving company performance [55].

As market competition becomes increasingly fierce, green innovation capability is becoming increasingly widely concerned by society. Companies' protection and attention to the environment have been strengthened by deepening their ESG practices. Although companies have gradually paid attention to carrying out ESG practices and focus on the sustainable development route of enterprises, there aren't many studies about how the ESG performance of companies affects corporate green innovation. A portion of the literature has focused solely on the social responsibility component, contending that social responsibility institutions and performance favorably influence the quantity and caliber of corporate green innovation [4, 56]. The ESG performance of a firm is not well represented by the social responsibility perspective, which is simply one component of that performance. Some scholars have pointed out the positive association of environmental, social, and governance practices on corporate green innovation from three ESG dimensions. However, the sample is heterogeneous and covers different research settings [57]. Even though there is research that specifically investigates the influence of ESG on green innovation using ESG rating data for Chinese business channels [58], the paper has the following shortcomings: On the one hand, their main regression using whether firms receive ratings as a quasi-natural test is not very plausible because the sample includes unrated firms and the financial and green performance of firms that receive ratings is naturally higher than those of non-rated firms. Green innovation output and quality are correspondingly higher. Hence their empirical models are highly endogenous and cannot be considered a quasi-natural experiment, and the grouping method is not clean. On the other hand, they also discuss the effect of ESG-specific ratings on firms' green innovation, with a much smaller sample size than the stated DID regression, and the small difference in ratings does not reflect the difference in the refinement of corporate ESG performance, which therefore does not support their conclusions.

It is significant to recognize that ESG performance can influence corporate innovations, specifically how it affects business performance, share price, and corporate value. Therefore, to understand how ESG performance affects corporate innovation activities, business performance, share price, and enterprise value, we must first understand how it affects those activities. Understanding the impact of ESG scores on corporate green innovation activities, the specific mechanisms, and the asymmetry of the impact in different circumstances is of utmost practical importance in the context of environmental pollution and resource depletion to realize green economic development and corporate green transformation. To empirically analyze the relationship between corporate ESG scores and corporate green innovation, as well as its role mechanism and moderating effect, we use the number of green patent applications and the number of green patent citations to measure corporate green innovation, build an empirical model using ESG scores from 2010 to 2019 and data on the quantity and quality of green innovation from 2011 to 2020. We also make an empirical model with a sample of listed Chinese companies in Shanghai and Shenzhen. Our findings confirm our research hypothesis by demonstrating a favorable relationship between company ESG performance and green innovation.

Meanwhile, corporate ESG scores promote corporate green innovation activities mainly through two paths: improving investment efficiency and improving political and business relations. In addition, the stronger the green attributes of firms, the stronger the ESG's contribution to green innovation performance; the stronger the black attributes of firms, the weaker the positive impact of ESG on green innovation performance. We use Bloomberg ESG Disclosure Scores published by Bloomberg as a proxy variable for ESG for the following reasons. On the one hand, the scores data is published by a non-Chinese organization. Thus, it is more independent in evaluating the ESG of enterprises. On the other hand, the variable is score data, which overcomes the original rating problems that are the non-refined and non-accurate evaluation of the ESG of enterprises.

The main contributions of our study are as follows. Firstly, to overcome the lack of refinement and precision of ESG performance measurement in previous studies [58], we use the ESG score variables instead of the original ESG ratings. Secondly, we tested the effectiveness of the ESG evaluation system from two perspectives of green innovation quantity and quality. Existing studies on ESG only analyze from three perspectives of corporate performance, stock price, and value ignoring the environmental and green attributes of ESG. Studies on green innovation explore from a quantitative standpoint ignoring the quality of green innovation. The ESG concept is better reflected in our study's green innovation elements, and the green innovation quality is better reflected in the quantitative indicators used to measure green innovation. Thirdly, we examine the possible mechanisms of ESG influence on green innovation and analyze the asymmetry of ESG influence on green innovation in terms of the green and black markets.

The remainder of the paper is as follows. Section 2 introduces the relevant theoretical foundations and then presents the relevant research hypotheses. Section 3 outlines the data selection and model design. Section 4 presents and analyzes the empirical results. Section 5 provides robustness tests. In Section 6, the final section, we conclude with a discussion.

## Theory and hypotheses development

### ESG and green innovation

Enterprises pay more attention to the relationship with corporate stakeholders, whether it is the green innovation activities based on technology or market-oriented business models [59], and put more emphasis on the creation of multiple integrated values based on innovation-led economic, social, and environmental [9] which are all associated with the ESG performance. Green innovation is a type of innovation where businesses try to use resources more efficiently and use less energy, and employ cutting-edge techniques to accomplish the twin objectives of economic and environmental performance [1]. Through green process product innovation [2, 60], businesses can reduce emissions and save energy. The advantages of companies' environmental, social, and governance practices favorably increase the intensity of green technology innovation [57], so the impact of ESG on corporate green innovation is mainly reflected in the following three aspects.

First of all, the environmental responsibility of enterprises contributes to the promotion of green innovation activities. Businesses' production and operation activities are under significant pressure from the legal limits of environmental rules and the informal constraints of public environmental expectations. Enterprises engage in environmental responsibility while compelled to take steps to enhance their environmental performance to preserve a positive environmental reputation. Consequently, for environmental performance, energy saving, and emission reduction, enterprises must use green innovation technology to improve production technology to achieve clean production. And financial institutions consider companies' environmental compliance when making investment and financing decisions. Therefore good environmental performance can alleviate financing constraints by reducing the financing cost of enterprises [61]. Furthermore, the environment is an essential external stakeholder for companies, and active corporate environmental responsibility helps to promote environmental cooperation. Companies are more likely to gain ideas about environmental management from their partners, including suppliers, to drive responsible green innovation projects [62].

Second, corporate social responsibility will promote green innovation by improving the relationship with stakeholders and providing them with the resources and information needed for green innovation activities. According to the stakeholder theory, actively undertaking social responsibility can assist companies in establishing broader and stronger relationships

with multiple stakeholders, such as customers, investors, regulators, and the public. These stakeholders will support enterprises' green innovation activities by increasing consumption and investment [63]. Based on the resource-based theory, corporate social responsibility is conducive to gaining the trust of stakeholders, including investors and consumers, and getting the market and financial resources needed for green innovation. Companies increase their green innovation investment, thus promoting green corporate innovation [30, 58]. According to signaling theory, on the one hand, CSR has the "information effect" of alleviating information asymmetry and principal-agent problems, thus providing information to help enterprises make long-term decisions on green innovation activities, which enables them to obtain more green patent outputs [64]. On the other hand, the active fulfillment of social responsibility can send positive signals to the market about the good business performance of the company, which indicates that the company is capable of participating in social responsibility activities with its resources and helps to attract positive media attention and improve their reputation and brand image [65, 66], and enterprises faceless media pressure to enhance their risk tolerance for innovation failure and stimulate their innovation energy, which in turn drives them to conduct green innovation activities with high uncertainty [67, 68]. In addition, higher participation in socially responsible activities enhances firms' product market recognition [69]. As green markets develop and consumer demand soars dramatically, firms are more willing to engage in environmentally friendly green innovation activities to increase corporate value.

Third, the better the internal governance, the higher the performance of corporate green innovation [70]. As green innovation activities have the characteristics of higher risk and longer cycle, enterprises do not tend to make innovation investment decisions, thus hindering green innovation, but good corporate governance alleviates principal-agent conflicts through incentive and constraint mechanisms, prompting corporate management to increase corporate R&D and innovation investment to achieve long-term sustainable corporate development [71, 72]. In addition, better internal governance can improve corporate performance, thus providing continuous and stable financial support for long-term corporate green innovation activities by mobilizing internal and external resources. Furthermore, ESG can promote corporate green innovation by optimizing corporate governance structures [73]. Gender diversity in the board of directors and executive management promotes more ESG practices in firms [74]. Board diversity can help companies become green organizations by promoting corporate ESG practices to stimulate green creativity, which drives companies to engage in green innovation [75].

Otherwise, ESG can help businesses adopt the ideas of sustainable development and creative growth [76]. On the one hand, these ideas encourage companies to pursue energy conservation, emission reduction, and clean production goals, as well as to increase their innovation spending and adopt technology that reduces energy consumption and protects the environment [77] to apply in their production and operation processes. On the other hand, enterprises with the guidance of green concepts invest their funds in green projects through green financing to achieve a pro-environmental and pro-climate transformation of internal capital flows or make investment funds further greener to provide strong support for green activities, including green innovation activities, which can encourage corporate innovation in green.

Accordingly, we obtained the research hypothesis:

H1: ESG is positively correlated with corporate green innovation.

## Mechanism of investment efficiency

The ESG evaluation system provides information and resources to support corporate investment and forms constraints on corporate investment. In general, the higher the ESG score, the

better performance of the company, the better the relationship between the firm and its stakeholders, and the more willing they are to provide information and resources to support the firm. At the same time, the ESG score also imposes constraints on enterprises, or to sustain the current score, enterprises have to make more green investments to maintain their corporate image, and the relevant investments are by the policy call and public expectation, so ESG enhances the efficiency of enterprises' investment through both resource support and investment constraints. In addition, it has been shown that corporate social responsibility can reduce agency costs and information asymmetry, so ESG firms have a low cost of equity [78]. The higher the ESG score of a firm, the lower the cost of equity, which is conducive to further enhancing the firm's investment efficiency. The higher the investment efficiency, the lower the inefficient investment, the closer to the optimal investment level, the more the resource utilization rate is about sufficient, the more the innovation output, and the more the green innovation output, that is, the more the green innovation patents obtained by the enterprise. Therefore, the higher the ESG score, the more efficient the firm's investment and green innovation output. Accordingly, we propose the research hypothesis:

H2: ESG can promote corporate green innovation by promoting investment efficiency.

## Mechanism of government-business relations

The higher the ESG score, the better the relationship between the company and its stakeholders. In Asia, many government-backed investment funds inject large amounts of money into ESG activities to reflect the importance of ESG practices for social development [79]. In China, companies pay particular attention to their relationship with the government because a good relationship with the government provides them with government support, such as government subsidies and tax breaks, and facilitates their financing, production, and management by obtaining government approval. In recent years, the local ecological environment has been related to the performance of the local government. Government regulation, technology push, and market pull are the three major influencing factors on carbon technology innovation activities. Government regulation is the only factor positively influencing carbon technology innovation activities [80]. The promotion of green technology innovation in China cannot be achieved without the power of the government, and the connection between the government and firms will impact enterprises' green technology innovation activities. Therefore, the better the ESG performance of a company, the more the government will support it, and conversely, its development will be restricted by the government. Since green innovation is long-term and risky [6], this greatly constrains the willingness and confidence of firms to make green innovation decisions. However, firms that maintain a good relationship with the government can gain more government support to share innovation risks and losses [34], encouraging firms to engage in green innovation activities. In summary, ESG scores can improve the relationship between government and firms, provide more resources for green innovation, and thus promote innovation. Therefore, we develop the following research hypothesis.

H3: ESG scores promote corporate green innovation through improving government-business relations.

## Moderation effect of green and black attributes

The ESG evaluation, one of the critical components of the green financial system, can contribute to green finance by promoting the effectiveness of financial resource allocation through the green flow of funds, thereby addressing the issue of environmental externality. This system primarily affects the financing of small and medium-sized businesses. And the companies internal and external environmental variables impact the green micro effect of the ESG system.

The stronger the black attribute, the stronger the environmental information asymmetry, the greater the environmental risk, and the more inclined enterprises are to make green bleaching behavior to cover up the poor environmental performance, thus maintaining the false ESG score and the external regulation will identify the green bleaching behavior of enterprises and thus inhibit the role of ESG. Nevertheless, when green attributes are stronger, environmental information asymmetry is smaller, and the environmental risks faced by firms are reduced, prompting ESG scores to be more objective to the ESG performance of firms, thus making full utilization of the green micro effects of the ESG system.

In summary, we obtain the following research hypotheses. As a green attribute of a company, an increase in environmental disclosure is conducive to promoting corporate ESG practices. Such environmental and ethical practices can promote the legitimization of corporate activities, improve corporate image and thus increase corporate financial performance [81]. Companies increase their investment in green technology innovation and enhance their innovation capabilities.

H4: Black attributes can weaken the positive effect of ESG scores on the green innovation of firms, and green attributes can enhance the promotion effect of ESG scores on the green innovation of a company.

## Methodology

### Sample and data

The sample of this study is a research sample of Chinese listed businesses in Shanghai and Shenzhen A-shares from 2010–2019 to analyze the impact of ESG on corporate green innovation performance. We conduct the following treatment for the sample: firstly, we remove the samples that were ST, PT, and *ST; secondly, we remove listed companies in the financial sector; thirdly, we remove companies listed before 2010; fourthly, we remove the samples with missing main variables. After processing, we finally obtained 8258 annual observations of 1090 listed companies. We use a 1% and 99% tail reduction (Winsorize) for the primary variables.

The data green on patents is from the China Research Data Service Platform. The data (CNRDS) on corporate finance is from the CSMAR and Wind databases, data on environmental disclosure from social responsibility reports published by Hexun.com, data on corporate ESG scores are from Bloomberg's Corporate Social Responsibility Disclosure Index (Bloomberg ESG Disclosure Scores), regional environmental data and economic data are from provincial statistical yearbooks, and macroeconomic data are from CEINet.

We declare that we have no human participants, human data, or human issues. We do not have any individual person's data in any form.

### Variables

**Explained variable.** The explanatory variables in this paper are corporate green innovation. We define firms' green innovation performance as quantitative and qualitative to obtain two explanatory variables for the number of green innovations (*GI*) and green patent citations (*GC*). The green patent is the most widely selected indicator to measure the green innovation ability of enterprises. The number of green patents granted reflects an enterprise's green innovation level more than the number of green patent applications, so we add one to the number of green patents granted and take the logarithm to measure the quantity of green innovation (*GI*) of enterprises. For the quality of green innovation, most existing scholars choose to measure the number of green invention patents and the number of green patents cited, among which the number of patents cited is more convincing than the invention patents [58], so in

this paper, we choose the number of green patents cited plus one and take the natural logarithm to measure the quality of green innovation (*GC*) of enterprises.

**Explanatory variable.** The core explanatory variable in this paper is the ESG score of firms. The ESG data is derived from the Bloomberg ESG Disclosure Scores, which consists of the ESG composite score and the ESG sub-scores with 122 sub-scores on 21 topics in three major categories.

**Intermediary variables.**

*(1) Efficacy of investments comes first (IE).* We utilize the absolute value of the residuals from the subsequent regression to measure inefficient investment as Model (1) [82]. The larger the indicator, the less efficient the firm's investment.

$$CI_{i,t} = \beta_0 + \beta_1 SG_{i,t-1} + \varepsilon_{i,t} \tag{1}$$

In Model (1), $CI_{i,t}$ represents the investment level of an enterprise, which is the proportion of fixed and intangible assets to total assets. $ESG_{i,t}$ represents the investment opportunity of an enterprise, which is the growth rate of sales revenue. The residual term represents the proportion of inefficient investment in the total investment, and the absolute value is taken to obtain the investment efficiency index *IE*. The larger the value, the less efficient the investment.

*(2) Government subsidies (Subsidy).* We use the normalized government subsidy (Subsidy) as a proxy variable for the government-enterprise relationship, which reflects the characteristics of the sample. The larger value indicates that means, the more government subsidy a firm receives, the better the relationship between the firm and the government

**Control variables.** By previous studies [4, 15, 56, 83], we take into account variables such as the firm's age (year of foundation), gearing (leverage), return on total assets (ROA), and Tobin's Q. (Q), net cash from investing activities (*ICF*), fixed assets (*Fix*), foreign ownership (*QFII*), dual employment (*Dual*), and audit opinion (*Opinion*). The key variables used in the empirical analysis are shown in Table 1.

## Model

**Baseline model.** Our data are short panel data, so a baseline regression model can represent the significant relationship between the independent variable ESG score and the dependent variable green innovation level. We use this model to control for year-fixed, industry-fixed, and province-fixed effects to control for the effect of unobservable factors at the industry and province levels overtime on the relationship between ESG score firms and green innovation level, and to city-level clustering. In addition, we can use the model to further examine the mechanisms and moderators of ESG scores affecting firms' green innovation. Based on the prior analysis and variable definitions, we use Model (2) for testing hypothesis H1.

$$GI_{i,t+1} = \alpha_0 + \alpha_1 ESG_{i,t} + \gamma X_{i,t} + \lambda_t + \eta_j + \varepsilon_{i,t} \tag{2}$$

Where $GI_{i,t+1}$ repents the firm i's level of green innovation in year t+1, $ESG_{i,t}$ denotes the firm *i*'s Bloomberg ESG score in that year, $X_{i,t}$ suggests a series of control variables, $\lambda_t$ denotes time fixed effects, $\eta_j$ denotes industry fixed effects; and $\varepsilon_{i,t}$ represents the random disturbance term.

**Intermediation model.** To test H2 and H3, the mediating effects of investment efficiency (*IE*) and government-enterprise relationship (*Subsidy)*, this paper further sets up the following mediation model and sets up the following testing steps [84, 85]. First, Model (1) shows the results of the regression model of corporate green innovation on ESG score. If $\beta_1$ is significant,

**Table 1. Descriptive statistics of the variables.**

| Variable classification | Variable name | Variable symbol | Variable definition |
|---|---|---|---|
| Explained variables | Quantity of Green Innovation | $GI_{t+1}$ | The logarithm of the number of green patents granted plus one to take the logarithm |
| | Quality of Green Innovation | $GCI_{t+1}$ | The logarithm of the number of green patent citations plus one to take the logarithm |
| Core explanatory variables | ESG Score | $ESG$ | The logarithm of Bloomberg ESG |
| Intermediate variables | Investment efficiency | $IE$ | Estimated from the Model (1) |
| | Government Grants | $Subsidy$ | Normalized government grants |
| Control variables | Years of Establishment | $Age$ | Ln(year—year of establishment) |
| | Gearing Ratio | $Leverage$ | Total liabilities/total assets |
| | Total Return on Assets | $ROA$ | Total profit/total assets |
| | Tobin's Q | $Q$ | Total market capitalization/total assets |
| | Net cash from investing activities | $ICF$ | Net cash from investing activities/total assets |
| | Fixed Assets | $Fix$ | Fixed Assets/Total Assets |
| | Foreign equity holdings | $QFII$ | Foreign shareholding ratio |
| | Two positions in one | $Dual$ | The value is 1 if the chairman is also the general manager; otherwise, it is 0 |
| | Audit opinion | $Opinion$ | The standard unqualified opinion takes the value of 1; otherwise, it is 0 |

the second step is carried out. Second, the regression equation of ESG score and mediating variables (*IE* and *Subsidy*) on corporate green innovation is constructed. The mediating mechanism exists if μ2 is significant and the signs of μ2 and β1 are the same.

$$IE_{i,t}/Subsidy_{i,t} = \beta_0 + \beta_1 ESG_{i,t} + \gamma X_{i,t} + \lambda_t + \eta_j + \varepsilon_{i,t} \quad (3)$$

$$GI_{i,t+1} = \mu_0 + \mu_1 ESG_{i,t} + \mu_2 IE_{i,t}/Subsidy_{i,t} + \gamma X_{i,t} + \lambda_t + \eta_j + \varepsilon_{i,t} \quad (4)$$

Where $IE_{i,t,}$ and $Subsidy_{i,t}$ represent the investment efficiency and government subsidies, respectively, and the rest of the variables are consistent with the baseline model.

**Moderating effect model.** To test H4, the moderating effect of the environmental attributes of firms, the following regression Model(5) was set up based on the baseline model.

$$GI_{i,t+1} = \alpha_0 + \alpha_1 ESG_{i,t} + \alpha_2 ESG_{i,t} \times R_{it} + \gamma X_{i,t} + \lambda_t + \eta_j + \varepsilon_{i,t} \quad (5)$$

Where *R* consists of the black and green attributes of the company. Black attributes include regional, industry, and company pollution attributes. We use the high pollution region dummy variable *HPP* (The regional pollution index for the current year takes a value of 1 if it is higher than the average value, and 0 otherwise.), the high pollution industry dummy variable *HPI* (high pollution industry takes a value of 1 otherwise it takes a value of 0) and the high pollution company dummy variable *HPC* (If the enterprise is a key pollution monitoring unit take the value of 1, otherwise it takes the value of 0) separately to measure black attributes. Green attributes include provincial, city, and firm environmental attributes. We employ provincial green finance *DGF* (normalized green finance index), city green innovation *DGI* (ratio of the total number of green patents in the city to the current year's average), and corporate environmental disclosure (the number of quantitative disclosures of environmental liability items as a proportion of the total number of items) as green attributes. And the remaining variables are consistent with Model 2.

**Table 2. Descriptive statistics of the main variables.**

| Variables | N | Mean | S. D. | Max | Median | Min |
|---|---|---|---|---|---|---|
| $GI_{t+1}$ | 8258 | 0.25 | 0.62 | 2.48 | 0.00 | 0.00 |
| $GC_{t+1}$ | 8258 | 0.41 | 0.92 | 4.56 | 0.00 | 0.00 |
| ESG | 8258 | 2.97 | 0.31 | 3.77 | 2.99 | 2.21 |
| E | 6950 | 2.16 | 0.67 | 3.72 | 2.23 | 0.73 |
| S | 8035 | 3.07 | 0.41 | 4.03 | 3.13 | 1.95 |
| G | 8258 | 3.80 | 0.11 | 4.05 | 3.80 | 3.52 |
| Age | 8258 | 2.87 | 0.32 | 3.53 | 2.89 | 1.61 |
| Leverage | 8258 | 0.47 | 0.20 | 0.89 | 0.48 | 0.05 |
| ROA | 8258 | 7.29 | 6.17 | 36.44 | 5.95 | -8.26 |
| Q | 8258 | 1.90 | 1.24 | 8.78 | 1.48 | 0.88 |
| ICF | 8258 | -0.06 | 0.08 | 0.17 | -0.05 | -0.39 |
| Fix | 8258 | 0.23 | 0.18 | 0.70 | 0.19 | 0.00 |
| QFII | 8258 | 0.17 | 0.54 | 2.79 | 0.00 | 0.00 |
| Dual | 8258 | 0.20 | 0.40 | 1.00 | 0.00 | 0.00 |
| Opinion | 8258 | 0.99 | 0.12 | 1.00 | 1.00 | 0.00 |

**Descriptive statistics.** The results of the descriptive statistics for the primary variables are shown in Table 2, where the mean value of green patents (*GI*) is 0.25, the standard deviation is 0.62, the maximum value is 0.48, and the minimum value is 0. This data suggests that the sample enterprises' average level of green innovation is low and that there is significant enterprise-level variation in their capacity for green innovation. ESG scores (*ESG*) vary significantly among businesses; the mean value is 2.97, the standard deviation is 0.31, the maximum is 3.77, the minimum is 2.21, and the median value is 2.99.

**Correlation test.** The Pearson correlation coefficient test matrix is displayed in Table 3. We can infer from Table 3 that there is a significant positive association between ESG score and corporate green innovation, which supports H1 preliminarily.

**Panel unit root test.** The existence of unit roots in panel data can have serious consequences, such as pseudo-regression, so we use both the Im-Pesaran-Shin test and Levin-Lin-Chu test to perform unit root tests to ensure the smoothness of each variable. Table 4 shows the results of the panel unit root tests. It can be seen that all variables are stationary at the 1% level, which means no unit root exists in the series. The results strongly reject the null

**Table 3. Pearson correlation coefficient test.**

| | $GI_{t+1}$ | $GC_{t+1}$ | ESG | E | S | G |
|---|---|---|---|---|---|---|
| $GI_{t+1}$ | 1.000 | | | | | |
| $GC_{t+1}$ | 0.513*** | 1.000 | | | | |
| ESG | 0.118*** | 0.200*** | 1.000 | | | |
| E | 0.110*** | 0.182*** | 0.833*** | 1.000 | | |
| S | 0.101*** | 0.154*** | 0.820*** | 0.508*** | 1.000 | |
| G | 0.028*** | 0.098*** | 0.514*** | 0.260*** | 0.306*** | 1.000 |

Note

***p < 0.01

**p < 0.05

*p < 0.1.

**Table 4. Panel unit root test.**

| Variables | Im-Pesaran-Shin test | | | Levin-Lin-Chu test | | |
|---|---|---|---|---|---|---|
| | t-bar | W[t-bar] | P-value | t-value | t-star | P-value |
| $GI_{t+1}$ | -1.814 | -6.819 | 0.000*** | -49.167 | -9.414 | 0.000*** |
| $GC_{t+1}$ | -1.898 | -8.581 | 0.000*** | -47.758 | -37.025 | 0.000*** |
| ESG | -1.657 | -3.540 | 0.000*** | -45.468 | -24.492 | 0.000*** |
| Age | -2.509 | -21.349 | 0.000*** | -71.791 | -65.239 | 0.000*** |
| Leverage | -1.850 | -7.574 | 0.000*** | -56.996 | -44.343 | 0.000*** |
| ROA | -2.044 | -11.631 | 0.000*** | -57.075 | -38.191 | 0.000*** |
| Q | -1.838 | -7.325 | 0.000*** | -55.392 | -33.445 | 0.000*** |
| ICF | -2.033 | -11.404 | 0.000*** | -50.308 | -28.543 | 0.000*** |
| Fix | -1.984 | -10.378 | 0.000*** | -57.179 | -41.668 | 0.000*** |
| QFII | -6.425 | -103.261 | 0.000*** | -161.793 | -164.627 | 0.000*** |
| Dual | -4.737 | -67.945 | 0.000*** | -391.658 | -415.238 | 0.000*** |
| Opinion | -1.988 | -10.455 | 0.000*** | -283.569 | -300.026 | 0.000*** |

hypothesis of unit root, so we can argue that the data are stable and there is no biased information in the panel.

## Empirical results and analysis

### Baseline regression

The ESG benchmark regression results are shown in Table 5. The explanatory variables in columns (1)-(4) are the quantity of green innovation. In column (1), the coefficient of *ESG* on the number of green innovation patents is 0.300, which is significant at the 1% level, indicating that ESG can increase the number of green innovation patents for companies. Based on the three sub-items of the ESG evaluation, we replace ESG with the natural logarithm of the corresponding scores for Environmental *E*, Social *S*, and Corporate Governance *G*. In column (2)-(4), the coefficient estimates of *E* and *S* are significantly positive at the 1% level, and the coefficient estimates of *G is* significantly positive at the 10% level, indicating that E, S, and G scores all promote the level of green innovation in companies. The explanatory variables in columns (5)-(8) are the quality of green innovation. In column (5), the regression coefficient of *ESG* is 0.610, which is significant at the 1% level, which suggests that ESG encourages business citation of green innovation patents. In columns (6)-(8), E, S, and G coefficient estimates are all significantly positive at the 1% level. The coefficient values are increasing in order, demonstrating that the positive effects of *E*, *S*, and *G* on the quality of green patents are in the order of G, S, E. The result above indicates that E, S, and G scores all promote the quality of green innovation in companies.

The regression results show that the amount and quality of green innovation output increase with increasing ESG score, supporting H1. In addition, our regression results indicate that all three subcategories of ESG can promote the quantity and quality of green innovation in enterprises. For the subscores of corporate ESG scores, we find that the E score has the most significant impact on corporate green innovation, and the G score has the least significant impact on corporate green innovation. Still, overall, the subscores of ESG all drive the quantity and quality of corporate green innovation. The descriptive statistics of the remaining control variables are generally consistent with existing studies [35, 55, 58].

The results illustrate that ESG scores can increase the quantity and quality of green innovation and that ESG is a sustainable "substantive innovation" rather than a "masked innovation"

**Table 5. Baseline regression results.**

| Variables | (1) | (2) | (3) | (4) | (5) | (6) | (7) | (8) |
|---|---|---|---|---|---|---|---|---|
| | GI_1 | GI_1 | GI_1 | GI_1 | GC_1 | GC_1 | GC_1 | GC_1 |
| ESG | 0.300*** | | | | 0.610*** | | | |
| | (7.04) | | | | (7.32) | | | |
| E | | 0.123*** | | | | 0.267*** | | |
| | | (6.68) | | | | (6.87) | | |
| S | | | 0.167*** | | | | 0.322*** | |
| | | | (5.75) | | | | (6.39) | |
| G | | | | 0.312* | | | | 0.981*** |
| | | | | (1.75) | | | | (3.11) |
| Age | -0.193** | -0.194** | -0.195** | -0.203** | -0.223 | -0.217 | -0.226 | -0.249 |
| | (-2.09) | (-2.00) | (-2.11) | (-2.27) | (-1.33) | (-1.12) | (-1.34) | (-1.52) |
| Leverage | 0.183** | 0.198** | 0.201*** | 0.209*** | 0.334** | 0.390** | 0.379** | 0.367** |
| | (2.35) | (1.98) | (2.66) | (2.83) | (2.16) | (2.01) | (2.56) | (2.40) |
| ROA | 0.002 | 0.002 | 0.003 | 0.003 | -0.006* | -0.007* | -0.005 | -0.006 |
| | (0.89) | (0.95) | (1.22) | (1.17) | (-1.78) | (-1.87) | (-1.54) | (-1.55) |
| Q | -0.028** | -0.037*** | -0.032*** | -0.034*** | 0.012 | -0.006 | 0.004 | 0.003 |
| | (-2.30) | (-2.62) | (-2.72) | (-2.86) | (0.68) | (-0.28) | (0.24) | (0.16) |
| ICF | -0.079 | -0.125 | -0.063 | -0.066 | -0.440* | -0.590** | -0.411* | -0.424* |
| | (-0.63) | (-0.78) | (-0.48) | (-0.50) | (-1.86) | (-1.99) | (-1.67) | (-1.71) |
| Fix | -0.035 | -0.069 | 0.018 | 0.035 | -0.387*** | -0.478*** | -0.290* | -0.260* |
| | (-0.26) | (-0.49) | (0.13) | (0.25) | (-2.74) | (-2.91) | (-1.92) | (-1.74) |
| QFII | 0.026 | 0.034 | 0.029 | 0.029 | 0.032 | 0.044 | 0.042 | 0.038 |
| | (0.83) | (1.08) | (0.91) | (0.90) | (0.75) | (0.98) | (0.95) | (0.86) |
| Dual | 0.041 | 0.047 | 0.035 | 0.029 | 0.096 | 0.103 | 0.077 | 0.076 |
| | (1.16) | (1.26) | (1.03) | (0.76) | (1.27) | (1.25) | (1.04) | (0.96) |
| Opinion | 0.081* | 0.087 | 0.079* | 0.102** | -0.123 | -0.080 | -0.122 | -0.087 |
| | (1.80) | (1.61) | (1.76) | (2.19) | (-1.18) | (-0.68) | (-1.14) | (-0.78) |
| Constant | -0.128 | 0.498 | 0.231 | -0.446 | -0.712 | 0.373 | 0.017 | -2.645* |
| | (-0.40) | (1.63) | (0.73) | (-0.55) | (-1.28) | (0.66) | (0.03) | (-1.80) |
| Y/I/P FE | YES | YES | YES | YES | YES | YES | YES | YES |
| Observations | 8,258 | 6,950 | 8,035 | 8,258 | 8,258 | 6,950 | 8,035 | 8,258 |
| Adj R$^2$ | 0.114 | 0.120 | 0.107 | 0.0975 | 0.0993 | 0.0997 | 0.0830 | 0.0754 |

Note: T-statistics calculated for city-level clusters in parentheses.

to simply whitewash financial statements. It is worth mentioning that the G score affects the number of green innovations less significantly than the E and S scores, probably because green innovation projects crowd out the firm's inherent resources and conflict with its short-term financial performance. We also find that when the explanatory variable is replaced with the number of green patents cited, all three aspects of ESG significantly improve the quality of green innovation at the 1% level. The coefficient of the G score is the largest. This result indicates that executives value the strategic perspective of the company's long-term development and choose to make high-quality green innovations to improve the company's competitiveness, so companies with good green strategies significantly improve the quality of green innovation.

Our results affirm the positive significance of ESG practices for green innovation, which positively affect companies' green transformation. The results also demonstrate the critical

role of the ESG scores of companies in influencing their green innovation decisions and that favorable practices in environmental, social, and governance aspects of companies will jointly promote corporate green innovation, achieve a sustainable development path for enterprises, and promote the integration of environmental, social and economic effects of enterprises.

### Intermediary mechanism analysis

**Mechanism of investment efficiency.**   The regression results for the mediating influence of investment efficiency are shown in columns (1)-(3) of Table 5. The coefficient estimate of *ESG* in column (1) is -3.183 and significantly negative at the 1% level, which means businesses with higher ESG scores make better investors. The coefficient estimates of $IE_{t+1}$ in columns (2) and (3) are -0.003 and -0.005, respectively, and are statistically negative at the 5% level, indicating the existence of this mediating effect and the relationship between investment efficiency and green innovation performance, At the 1% level, both of the coefficient estimates of *ESG* in columns (2) and (3)—0.291 and 0.594, respectively—are significantly positive. Both columns (2) and (3) coefficient values of *ESG* 0.291 and 0.594, respectively—are statistically significant at the 1% level. The regression results suggest that ESG performance contributes to green innovation by improving firms' investment efficiency. As a result, H3 should be accepted.

The results suggest that the fulfillment of ESG responsibilities will drive companies to make green investments to cater to investors' preference for environmentally friendly companies and that ESG practices are conducive to improving the efficiency of investments and the utilization of internal and external resources, which in turn will make companies willing to engage in more green innovation activities and improve their green technological innovation capabilities.

**Mechanism of government-business relations.**   Columns (4) to (6) of Table 6 show the regression results for the mediating effect of the government-firm relationship. The coefficient estimate of *ESG* in column (4) is 0.027 and significantly positive at the 1% level, indicating that the higher the ESG score, the more government subsidies the firm receives. In other words, the ESG score significantly improves the relationship between the government and the firm; the coefficient estimates of *Subsid*y in columns (5) and (6) are respectively 0.2289 and 5.407, and both are positive at the 1% level, which means that government subsidies significantly promote green innovation, so the better the relationship with the government, the more government subsidies the enterprises receive, and the more funds they have to engage in green

**Table 6. Regression results for mediating mechanisms.**

| Variables | (1) | (2) | (3) | (4) | (5) | (6) |
|---|---|---|---|---|---|---|
| | $IE_{t+1}$ | $GI_{t+1}$ | $GC_{t+1}$ | Subsidy | $GI_{t+1}$ | $GC_{t+1}$ |
| ESG | -3.184*** | 0.291*** | 0.594*** | 0.027*** | 0.239*** | 0.465*** |
| | (-4.65) | (7.16) | (7.33) | (3.23) | (5.14) | (5.50) |
| $IE_{t+1}$ | | -0.003** | -0.005** | | | |
| | | (-2.53) | (-2.38) | | | |
| Subsidy | | | | | 2.289*** | 5.407*** |
| | | | | | (10.80) | (7.58) |
| Constant | 13.907*** | -0.109 | -0.673 | -0.036* | -0.065 | -0.541 |
| | (3.50) | (-0.34) | (-1.25) | (-1.69) | (-0.21) | (-1.04) |
| Controls | YES | YES | YES | YES | YES | YES |
| Y/I/P FE | YES | YES | YES | YES | YES | YES |
| Observations | 8,258 | 8,258 | 8,258 | 8,258 | 8,258 | 8,258 |
| Adj R$^2$ | 0.082 | 0.139 | 0.163 | 0.252 | 0.115 | 0.101 |

innovation behavior. Thus this mediating effect exists. The coefficient value decreases compared to the baseline regression. Therefore the mediation effect is partial. The above regression results suggest that ESG performance promotes corporate green innovation by improving the relationship between government and business, which supports hypothesis H2.

The results indicate the important role of government-business relationships in mediating the impact of ESG performance on corporate green innovation. The government encourages and supports ESG practice projects, so companies that participate in ESG practice projects can build a good corporate image, maintain good relations with the government, and gain political resources, including government subsidies and the economic resources they bring. These competitive resources can be regarded as a kind of external risk protection, which can reduce the cost of green innovation and reduce the risk of R&D, and enhance the motivation of enterprises to invest in green innovation projects.

## Moderation effects analysis

The regression results of Panel A in Table 7 show that the interaction coefficients of *ESG* with *HPP*, *HPI*, and *HPC* decrease in significance and coefficient values compared with the estimated values of the baseline regression *ESG*, which indicates that the stronger the black attributes of the firm, the weaker the promotion effect of *ESG* on green innovation. The regression

**Table 7. Regression results for moderating effects of black and green attributes.**

| Variables | (1) | (2) | (3) | (4) | (5) | (6) |
|---|---|---|---|---|---|---|
| | $GI_{t+1}$ | $GC_{t+1}$ | $GI_{t+1}$ | $GC_{t+1}$ | $GI_{t+1}$ | $GC_{t+1}$ |
| **Panel A: Black Features** | | | | | | |
| ESG×HPP | 0.026 | 0.101** | | | | |
| | (1.40) | (2.54) | | | | |
| ESG×HPI | | | 0.015* | 0.038** | | |
| | | | (1.78) | (2.24) | | |
| ESG×HPC | | | | | 0.011 | 0.067*** |
| | | | | | (1.43) | (4.14) |
| Constant | 0.693** | 0.977** | 0.677** | 0.930* | 0.698** | 1.024** |
| | (2.40) | (1.99) | (2.31) | (1.86) | (2.38) | (2.05) |
| Controls | YES | YES | YES | YES | YES | YES |
| Y/I/P FE | YES | YES | YES | YES | YES | YES |
| Observations | 8,258 | 8,258 | 8,258 | 8,258 | 8,258 | 8,258 |
| Adj R² | 0.095 | 0.068 | 0.096 | 0.068 | 0.095 | 0.072 |
| **Panel B: Green Features** | | | | | | |
| ESG×DGF | 0.386*** | 0.826*** | | | | |
| | (3.99) | (3.83) | | | | |
| ESG×CGI | | | 0.572* | 1.306** | | |
| | | | (1.93) | (2.48) | | |
| ESG×EDG | | | | | 0.856*** | 1.360*** |
| | | | | | (4.80) | (4.31) |
| Constant | -0.469* | -1.521*** | 0.820** | 1.042** | 0.673** | 0.933* |
| | (-1.66) | (-2.64) | (2.58) | (2.13) | (2.34) | (1.91) |
| Controls | YES | YES | YES | YES | YES | YES |
| Y/I/P FE | YES | YES | YES | YES | YES | YES |
| Observations | 8,258 | 8,258 | 5,439 | 5,439 | 8,258 | 8,258 |
| Adj R² | 0.104 | 0.083 | 0.127 | 0.125 | 0.010 | 0.071 |

results of Panel B show that the coefficient values of *ESG* and *DGF*, *CGI*, and *EDG* are all significant at the 1% level and are greater than the baseline regression coefficient values, which suggests that the stronger the green attributes of firms, the greater the positive impact of *ESG* on green innovation.

The results demonstrate the opposed effects of corporate green and black attributes on the relationship between ESG scores and corporate green innovation. At the black attribute level, the sample, whether a highly polluting industry or a highly polluting firm, exacerbates environmental information asymmetry and exposes firms to higher environmental risks. Firms will mask the inherent risks through green bleaching practices. Thus ESG scores are more biased towards a false reflection of ESG performance and will weaken the positive effect of ESG scores on green innovation. At the level of green attributes, whether at the province, city, or firm level, green attributes can reduce environmental information asymmetry, make ESG scores more realistic and reliable reflections of firms' true ESG performance, and enhance the effectiveness of ESG scores in promoting corporate green innovation. The government should increase the punishment for polluting enterprises, increase the cost of polluting enterprises through environmental regulation pressure, and consciously promote the transformation of enterprises from black attributes to green attributes. And enterprises should increase the disclosure of environmental information to reduce the uncertainty of environmental information and enhance their green attributes, and at the same time, reduce emissions and environmental pollution by improving production processes and greening production to reduce their black attributes, to better utilize the positive effect of ESG performance on green innovation.

## Robustness tests

### Replacing measures of core variables

In the robustness test section, we use the number of green patent applications to measure the quantity of green innovation of the firm (*GGI_1*) and the number of green invention patents to measure the quality of green innovation of the firm (*INNO_1*). Table 8 reports the regression results for replacing the core variable measures. The results are consistent with the benchmark regression, where both the composite corporate ESG score and sub-scores contribute to the quantity and quality of corporate green innovation.

Table 8. Regression results for replacing core variables.

| Variables | (1) | (2) | (3) | (4) | (5) | (6) | (7) | (8) |
|---|---|---|---|---|---|---|---|---|
| | *GGI_1* | *GGI_1* | *GGI_1* | *GGI_1* | *INNO_1* | *INNO_1* | *INNO_1* | *INNO_1* |
| *ESG* | 0.389*** | | | | 0.264*** | | | |
| | (6.02) | | | | (5.48) | | | |
| *E* | | 0.186*** | | | | 0.122*** | | |
| | | (6.28) | | | | (4.76) | | |
| *S* | | | 0.187*** | | | | 0.153*** | |
| | | | (4.16) | | | | (4.40) | |
| *G* | | | | 0.326* | | | | 0.472** |
| | | | | (1.82) | | | | (2.26) |
| Constant | -1.058*** | -0.345 | -0.530 | -1.179 | -0.143 | 0.368 | 0.158 | -1.154 |
| | (-3.25) | (-1.06) | (-1.59) | (-1.45) | (-0.42) | (1.29) | (0.49) | (-1.19) |
| Controls | YES | YES | YES | YES | YES | YES | YES | YES |
| Y/I/P FE | YES | YES | YES | YES | YES | YES | YES | YES |
| Observations | 6,891 | 5,724 | 6,675 | 6,891 | 8,258 | 6,950 | 8,035 | 8,258 |
| Adj R$^2$ | 0.165 | 0.169 | 0.153 | 0.150 | 0.0741 | 0.0803 | 0.0682 | 0.0616 |

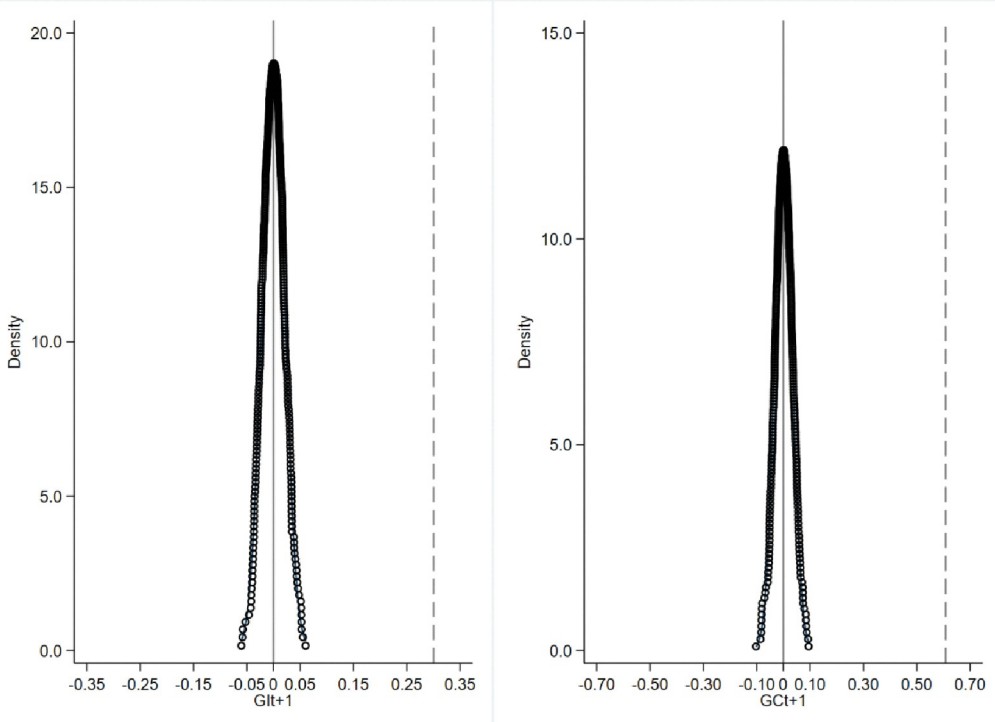

**Fig 1. Placebo test.**

## Placebo test

We used a non-parametric permutation test to perform a placebo test on the baseline regression. The placebo test is illustrated in Fig 1. We find that from the test results that the distribution of the estimated coefficients for the 500 random samples is close to a normal distribution with mean zero and that the coefficients of the benchmark regressions for $GI_{t+1}$ and $GC_{t+1}$ green innovation indicated by the dashed lines in the figure, Table 5 columns (3) and (6) are different from the correlation coefficients obtained from the non-parametric tests. Therefore, the test results exclude the possibility that the effect of ESG on green innovation performance is dependent on other unobservable factors. In other words, the interference of other events in the benchmark regression is excluded, and the obtained benchmark regression results are robust.

## Adding variables

We next control provincial and national level economic variables that may affect firms' green innovation based on the baseline regression column (1) to verify the robustness of the baseline regression results. We specifically introduce regional per capita gross product (*PerGDP*, the logarithm of regional per capita gross product), regional financial development level (*FD*, regional deposit and loan as a share of GNP), regional pollution level (*DPG*, industrial pollution investment as a share of GNP), broad money growth rate ($M_2$), and Shanghai interbank lending rate (*Shibor*, the annual 10-year Shanghai interbank lending average interest rate) to control for regional economic, environmental and macroeconomic effects on the benchmark regressions. Table 9 shows the results of the regressions with the addition of control variables. From the regression results, the coefficient estimates for ESG are all significantly positive at the 1% level. The regression results are generally consistent with the benchmark regression, which means the robustness of the benchmark regression.

**Table 9. Regression results for adding control variables.**

| Variables | (1) | (2) | (3) | (4) |
|---|---|---|---|---|
| | $GI_{t+1}$ | $GI_{t+1}$ | $GC_{t+1}$ | $GC_{t+1}$ |
| ESG | 0.301*** | 0.301*** | 0.609*** | 0.609*** |
| | (7.10) | (7.10) | (7.33) | (7.33) |
| FD | 0.002 | 0.002 | -0.002 | -0.002 |
| | (1.07) | (1.07) | (-1.21) | (-1.21) |
| DPG | -0.208** | -0.208** | 0.089 | 0.089 |
| | (-2.56) | (-2.56) | (0.59) | (0.59) |
| $M_2$ | | 0.288*** | | -0.142* |
| | | (5.05) | | (-1.72) |
| Shibor | | 1.043*** | | -0.476 |
| | | (4.90) | | (-1.55) |
| Constant | -5.300*** | -10.082*** | -0.341 | 1.829 |
| | (-2.73) | (-3.48) | (-0.13) | (0.46) |
| Controls | YES | YES | YES | YES |
| Y/I/P FE | YES | YES | YES | YES |
| Observations | 8,258 | 8,258 | 8,258 | 8,258 |
| Adj R$^2$ | 0.114 | 0.114 | 0.074 | 0.074 |

## Replacement regression models

$GI_{t+1}$ and $GC_{t+1}$ are discrete variables suitable for Poisson, Tobin, and Negative Binomial regression models. Table 10 shows the results of the substitution regression model. From the regression results, the coefficient estimates of *ESG* are all significantly positive at the 1% level. The regression results are generally consistent with the baseline regression, which suggests the robustness of the baseline regression.

## Instrumental variables approach

Using green innovation indicators for period t+1 avoids the problems associated with certain simultaneity biases while reducing the estimation error associated with reverse causality issues. However, the relationship between ESG and green innovation is still strongly endogenous, which means that firms with higher green innovation performance also have higher ESG scores. There may also be omitted variables that affect ESG scores. At the same time, there is

**Table 10. Regression results of the replacement model.**

| Variables | (1) | (2) | (3) | (4) | (5) | (6) |
|---|---|---|---|---|---|---|
| | | $GI_{t+1}$ | | | $GC_{t+1}$ | |
| | Poisson | Tobit | NB | Poisson | Tobit | NB |
| ESG | 1.181*** | 0.300*** | 1.169*** | 1.376*** | 0.609*** | 1.378*** |
| | (8.24) | (7.10) | (7.35) | (9.37) | (7.36) | (8.08) |
| Constant | -4.124*** | -0.148 | -3.901*** | -4.029*** | -0.737 | -3.857*** |
| | (-3.71) | (-0.46) | (-3.41) | (-3.72) | (-1.33) | (-3.43) |
| Controls | YES | YES | YES | YES | YES | YES |
| Y/I/P FE | YES | YES | YES | YES | YES | YES |
| Observations | 8,258 | 8,258 | 8,258 | 8,258 | 8,258 | 8,258 |
| Loglikelihood | -4437 | -7176 | -4338 | -6926 | -10599 | -6345 |
| Pseudo R$^2$ | 0.186 | 0.068 | 0.147 | 0.118 | 0.042 | 0.067 |

**Table 11. 2SLS and GMM results for instrumental variables.**

| Variables | (1) | (2) | (3) | (4) | (5) |
|---|---|---|---|---|---|
| | ESG | 2SLS $GI_{t+1}$ | $GC_{t+1}$ | $GI_{t+1}$ | $GC_{t+1}$ |
| | | | | GMM | |
| $ESGMean_{t-1}$ | 0.679*** | | | | |
| | (17.29) | | | | |
| ESG | | 0.757*** | 1.156*** | 0.755*** | 1.159*** |
| | | (6.89) | (6.56) | (6.87) | (6.58) |
| Constant | 2.766*** | -1.419*** | -2.260*** | -1.391*** | -2.238*** |
| | (54.35) | (-4.45) | (-4.45) | (-4.37) | (-4.40) |
| Observations | 8,258 | 8,258 | 8,258 | 8,258 | 8,258 |
| Controls | YES | YES | YES | YES | YES |
| Y/I/P FE | YES | YES | YES | YES | YES |
| Adj $R^2$ | 0.211 | 0.071 | 0.072 | 0.071 | 0.072 |
| F statistics | 35.95 | 19.64 | 12.26 | | |
| Kleibergen-Paaprk LM statistic | | 235.653 | 235.653 | | |
| Cragg-Donald Wald F-statistic | | 262.190 | 262.190 | | |
| Kleibergen-Paaprk Wald F-statistic | | 299.097 | 299.097 | | |

simultaneously an impact on firms' green innovation that makes the benchmark regressions biased and inconsistent. We use an instrumental variable to address this issue to eliminate the effect of potential endogeneity on the benchmark regression. This paper chooses the industry-level mean of ESG ($ESGMean_{t-1}$) of the previous year as the instrumental variable [84]. The industry influences the ESG score, but the industry-level mean is not directly related to the green performance of individual firms, so $ESGMean_{t-1}$ meets the requirements of an instrumental variable.

Before conducting the least squares regression of the instrumental variables, we first conducted a correlation coefficient test between $ESGMean_{t-1}$ and ESG. The Pearson correlation coefficient test results showed that the correlation coefficient between the two was 0.194 and significant at the 1% level, so we can initially conclude that the higher the industry ESG means, the higher the ESG performance of the firm. The outcomes of the 2SLS and GMM results for the instrumental variables are shown in Table 11. The first three columns are the estimated results of 2SLS. According to the regression results, the first stage's coefficient estimates of $ESGMean_{t-1}$ is 0.679 and significant at the 1% level, suggesting that the industry in which a company operates impacts its ESG performance. The second stage regression shows that the predicted ESG coefficients are considerably positive at the 1% level, demonstrating that ESG improves business performance regarding green innovation. After conducting the main regression, we conduct a series of tests for instrumental variables such as homogeneity of instrumental variables, weak instrumental variables, and over-identification, whose results show that the Model passes all tests. The last two columns are the estimated results of GMM. The regression results also validate the baseline hypothesis of this paper.

## Propensity score matching

To address the problem of sample selection bias, we choose the propensity score matching method (PSM), using a 1:1 nearest neighbor matching with a matching radius of 0.05, with whether it is a highly polluting industry as the grouping variable and all the control variables in column (1) as covariates, inducing age of establishment (Age), gearing (Leverage), return on total assets (ROA), Tobin's Q (Q), net cash from investing activities (ICF), fixed assets (Fix),

**Table 12. PSM-benchmark regression results.**

| Variables | (1) | (2) | (3) | (4) | (5) | (6) | (7) | (8) |
|---|---|---|---|---|---|---|---|---|
| | $GI_{t+1}$ | $GI_{t+1}$ | $GI_{t+1}$ | $GI_{t+1}$ | $GC_{t+1}$ | $GC_{t+1}$ | $GC_{t+1}$ | $GC_{t+1}$ |
| ESG | 0.260*** | | | | 0.504*** | | | |
| | (4.92) | | | | (5.98) | | | |
| E | | 0.099*** | | | | 0.236*** | | |
| | | (4.06) | | | | (4.89) | | |
| S | | | 0.138*** | | | | 0.260*** | |
| | | | (3.85) | | | | (6.10) | |
| G | | | | 0.284 | | | | 0.456 |
| | | | | (1.57) | | | | (1.05) |
| Constant | 0.016 | 0.663*** | 0.317 | -0.361 | -0.129 | 0.867* | 0.460 | -0.509 |
| | (0.05) | (2.69) | (1.16) | (-0.45) | (-0.23) | (1.84) | (0.96) | (-0.27) |
| Y/I/P FE | YES | YES | YES | YES | YES | YES | YES | YES |
| Observations | 3,379 | 2,894 | 3,274 | 3,379 | 3,379 | 2,894 | 3,274 | 3,379 |
| Adj R$^2$ | 0.109 | 0.111 | 0.103 | 0.098 | 0.082 | 0.080 | 0.072 | 0.069 |

foreign ownership (*QFII*), dual employment (*Dual*) and audit opinion (*Opinion*). After passing the common support hypothesis and parallel trend hypothesis tests, the benchmark regression was re-run, and the regression results are shown in Table 12. From the results, we find that the regression coefficients of *ESG* are all significantly positive at the 1% level. Meanwhile, the coefficient estimates of *E* and *S* are both significantly positive at the 1% level, but the coefficient estimate of G is not statistically significant after eliminating the problem of the sample, which indicates that the short-term corporate governance objectives of the company are contrary to the long-term green innovation activities, consistent with economic theory and experience.

## Conclusion and discussion

Green innovation is a crucial manifestation of corporate applying the ESG concept, which reflects the micro-green effect of the ESG evaluation system. Using panel data and the sample of Chinese listed businesses from 2010 to 2019, we empirically explore the impact of ESG scores on corporate green innovation from corporate investment efficiency and government-enterprise relations perspectives. The results indicate both the composite and sub-scores of a company's ESG contribute to the quantity and quality of its green innovation. And ESG supports corporate green innovation by increasing businesses' investment effectiveness and improving their government-business relationship. The results also show that corporate green attributes strengthen the promotion function of ESG on corporate green innovation. In contrast, black attributes reduce the beneficial effects of ESG on corporate green innovation.

According to our research, the following recommendations can be made for enhancing the ESG evaluation system and encouraging the sustainable growth of micro-enterprises. Firms need to implement the ESG concept, manage the various environmental risks they face, increase their level of pro-environmental preference, enhance the environmental disclosure mechanism, pay more attention to the non-financial performance of green performance, and promote business development and green development. The findings of this paper prove the importance of practicing environmental, social, and governance responsibilities and the positive significance of ESG performance for enterprises' green and sustainable development. The implementation of the ESG concept by enterprises is conducive to promoting the integration of environmental, social, and economic performance and achieving a win-win situation of environmental, social, and economic effects. Moreover, the findings of this paper also point

out the way and direction for enterprises to promote green innovation development. By actively fulfilling environmental and social responsibilities, enterprises can win the trust of stakeholders, including governments and investors, obtain key political and economic resources that are indispensable for green innovation, alleviate financing constraints, improve resource utilization, enhance the output and quality of green technology innovation, and embark on a sustainable green development transformation path.

Moreover, the government should implement green development into practice, create fiscal policies for businesses based on the ESG evaluation system, subsidize green enterprises and restrict black enterprises, and encourage businesses to engage in green innovation activities that adhere to ESG standards. The conclusion of this paper proves the key role played by good political-business relations between ESG scores and corporate green innovation. Therefore, the government should focus on the role of important political and economic resources, including government subsidies and tax incentives, and strongly support enterprises to carry out ESG practice projects that are beneficial to social development and progress to attract enterprises to participate in green innovation activities consciously and actively, thus guiding more enterprises to take the green development path.

Regulators should create distinct regulatory policies based on businesses' environmental risks and enhance the mechanism for exchanging environmental information to encourage companies to engage in green innovation activities. Regulators should pay attention to the environmental information disclosure of enterprises, timely detect the possible "greenwashing" behavior of enterprises and punish these enterprises, to promote the ESG score to reflect the ESG performance of enterprises more truly and let the ESG performance promote the green innovation of enterprises in practice, that is, let the green attributes better promote the positive link between ESG score and green innovation of enterprises, and weaken the inhibiting effect of black attributes on the relationship between the two.

Institutional investors need to pay attention to the ESG performance of enterprises and further incorporate ESG factors into their investment strategies to better identify enterprises' internal and external environmental risks and provide enterprises with corresponding funds based on ESG evaluation. As an important external stakeholder of enterprises, enterprises will pay attention to the investment tendency of institutional investors to obtain more financing support. Therefore, institutional investors pay attention to ESG investment concepts, environmental protection of enterprises, and sustainable development strategies, which are conducive to guiding enterprises to pay attention to ESG practices, fulfilling environmental and social responsibilities, and enhancing their green innovation drive.

The limitations of this paper lie in the following two aspects. On the one hand, we only explore the micro-green effect of the ESG evaluation system and do not analyze the role of the ESG evaluation system comprehensively. On the other hand, we ignored the motives of corporate greenwashing and failed to eliminate the part of corporate greenwashing in green innovation. Future research can examine the relationship between ESG scores and green innovation from two aspects. First, the research can analyze the role of ESG in greenwashing behaviors such as environmental performance, production performance, and investment efficiency. Second, future research will have indicators to identify green innovation drifting green motives to better examine the effectiveness of the ESG evaluation system.

## Author Contributions

**Conceptualization:** Danni Chen.

**Data curation:** Chunlian Zhang, Danni Chen.

**Formal analysis:** Chunlian Zhang, Danni Chen.

**Methodology:** Chunlian Zhang.

**Software:** Danni Chen.

**Supervision:** Danni Chen.

**Validation:** Chunlian Zhang, Danni Chen.

**Visualization:** Danni Chen.

**Writing – original draft:** Chunlian Zhang, Danni Chen.

**Writing – review & editing:** Chunlian Zhang.

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
