## [Decision Letter · Decision Letter 0]

27 Feb 2023

PONE-D-22-33103Do Corporate ESG Scores Improve Green Innovation?Empirical Evidence from Chinese Listed CompaniesPLOS ONE

Dear Dr. Chen,

Thank you for submitting your manuscript to PLOS ONE. After careful consideration, we feel that it has merit but does not fully meet PLOS ONE’s publication criteria as it currently stands. Therefore, we invite you to submit a revised version of the manuscript that addresses the points raised during the review process.

I agree with the reviewers that you paper is interesting in terms of novelty but it needs some improvements. In adittion to the reviewers' comments I encourage the authors to improve the discussion section. The paper must contribute to the literature and create new knowledge. The contributions are weak. Moreover, you need to take into account the gap you aim to fill in terms of the previous empirical evidence. In which way your findings shed light on previous studies, which some times could be confusing.I am concerned about the potential endogeneity issue in your paper. Either you justify, with references in the literature, the instrument you used or you could also apply GMM.You could also justify the 1:1 matching or use different machings, to improve the robustness of this technique.It could also advisable to improve the robustness of the dependent variable, since some studies show the sensitiveness of the results to the ESG variables.   In this way, I believe that you could use different ESG variables, such as ESG for controversies of the different ESG pillars which can be obtained from the ESG breakdown. 

We look forward to receiving your revised manuscript.

Kind regards,

José Antonio Clemente Almendros, PhD

Academic Editor

PLOS ONE

Journal Requirements:

"Funding The authors acknowledge the financial support from: the project of the Water Economy and Water Rights Research Center , a school-level platform in Nanchang Institute of Technology :“An empirical study on the Microeconomics of ESG performance under the ‘Dual-carbon’ vision (22ZXZD01)."       

Reviewers' comments:

Reviewer's Responses to Questions

**Comments to the Author**

1. Is the manuscript technically sound, and do the data support the conclusions?

Reviewer #1: Yes

Reviewer #2: Yes

2. Has the statistical analysis been performed appropriately and rigorously? 

Reviewer #1: Yes

Reviewer #2: Yes

3. Have the authors made all data underlying the findings in their manuscript fully available?

Reviewer #1: Yes

Reviewer #2: Yes

4. Is the manuscript presented in an intelligible fashion and written in standard English?

Reviewer #1: No

Reviewer #2: Yes

5. Review Comments to the Author

Reviewer #1: Dear authors, the paper wants to investigate the impact of the ESG scores on corporate green innovation performance, the specific mechanism of this effect, and the asymmetry of this impact under different moderation effects by

using Chinese listed A-shares in Shanghai and Shenzhen. The goal is specific and it required a very strong analysis.

Reading the paper, I', fashinating about the rigor in the strucutre and in the analysis process. So, congratulations for this.

But, some minor changes are required:

1. to realize a professional proofreading;

2. to improve the final part with some interesting scientific and managerial implications,

3. to improve the literature review with some papers (see below)

REFERENCES

Auer, B.R., Schuhmacher, F., Do socially (ir)responsible investments pay? New evidence from international ESG data, (2016) Quarterly Review of Economics and Finance, 59, pp. 51-62

ESG and corporate financial performance: the mediating role of green innovation: UK common law versus Germany civil law, Chouaibi, S., Chouaibi, J., Rossi, M. EuroMed Journal of Business, 2022, 17(1), pp. 46–71

Exploring the moderating role of social and ethical practices in the relationship between environmental disclosure and financial performance: evidence from esg companies, Chouaibi, S., Rossi, M., Siggia, D., Chouaibi, J. Sustainability (Switzerland), 2022, 14(1), 209

Bi, K., Huang, P., Wang, X. Innovation performance and influencing factors of low-carbon technological innovation under the global value chain: A case of Chinese manufacturing industry, (2016) Technological Forecasting and Social Change, 111, pp. 275-284.

The effect of corporate social responsibility and the executive compensation on implicit cost of equity: Evidence from French ESG data, Chouaibi, Y., Rossi, M., Zouari, G., Sustainability (Switzerland), 2021, 13(20), 11510

Reviewer #2: I thoroughly read the manuscript entitled "Do Corporate ESG Scores Improve Green Innovation? Empirical Evidence from Chinese Listed Companies" which I found interesting in term of title and novelty but before to published the paper please incorporates my suggestion and comments.

1. Please do not use word abbreviations in the topic. Please write the full word "ESG".

2. The introduction is too weak in term of latest study. Please write more clearly about the mechanism that how ESG score improve green innovation.

3. Please include some latest literature

4. I am really astonished why the authors do not explain the base line method? please explain the details. Foor example, why we are using this methodology? advantages? comparing to another methodology.

5. The method which authors used in this paper is too much old but its ok to calculate the statics of the variables. Instead, authors can use CS-ARDL (which can calculate short and long run) Quantile and Quantile Approach, Quantile via movement.

6. Secondly author should use second generation panel unit root, otherwise the results will be called spurious regression. Most importantly authors should use cross-sectional dependence (CD) test (Most of economies and provinces are linked with trade relations, globalization, technology, knowledge transfer, and financial capital flows).

7. Please the discuss the results in more good way like why its negative, positive and significant? what is Economics behind is that?

8. Please check the English. Gramar and sentence.

Best of Luck

6. PLOS authors have the option to publish the peer review history of their article (what does this mean?). If published, this will include your full peer review and any attached files.

Reviewer #1: **Yes: **Matteo Rossi

Reviewer #2: **Yes: **Faheem Ur Rehman

---

## [Author Response · Author response to Decision Letter 0]

12 Apr 2023

Dear Editors and Reviewers:

Thank you for your letter and for the reviewers' comments concerning our manuscript entitled "Do Corporate ESG Scores Improve Green Innovation? Empirical Evidence from Chinese Listed Companies" (PONE-D-22-33103). Those comments are all valuable and helpful for revising and improving our paper. We are very sorry to update the revised manuscript so late because of the addition of some necessary content. 

Note: In the response letter, all changes are marked in red. What's more, additions and important changes are marked in red and changes are marked in blue on the new manuscript . Also, The main corrections in the paper and the responds to the reviewer's comments are flowing:

I received comments from the reviewers, and they were valuable. Overall, I have made several improvements in the following areas.

Firstly, we have improved the discussion in my new manuscript. We have also added a lot of new literature to supplement the theoretical section.

Secondly, we have used an instrumental variable to address this issue to eliminate the effect of potential endogeneity on the benchmark regression. This paper chooses the industry-level mean of ESG of the previous year as the instrumental variable. The industry influences the ESG score, but the industry-level mean is not directly related to the green performance of individual firms, so ESGMeant-1 meets the requirements of an instrumental variable. The outcomes of the 2SLS and GMM results for the instrumental variables are shown in Table 11. 

Thirdly, we have discussed the role of the ESG composite score and the three pillar scores on the quantity and quality of green innovation. It is worth mentioning that the G score affects the number of green innovations less significantly than the E and S scores, probably because green innovation projects crowd out the firm's inherent resources and conflict with its short-term financial performance. We also find that when the explanatory variable is replaced with the number of green patents cited, all three aspects of ESG significantly improve the quality of green innovation at the 1% level. This result indicates that executives value the strategic perspective of the company's long-term development and choose to make high-quality green innovations to improve the company's competitiveness, so companies with good green strategies significantly improve the quality of green innovation. 

Response to Reviewer 1 Comments

Point 1: Realize a professional proofreading.

Response 1: We thank the reviewer for raising this question. Following the journal's requirements, we have carefully revised the paper format and corrected the grammatical errors.

Point 2: Improve the final part with some interesting scientific and managerial implications.

Response 2: According to your comments, we have revised the conclusions and suggestions of this paper in detail, making the system conclusions of this paper more rich and complete. 

As for the companies, implementing the ESG concept by enterprises is conducive to promoting the integration of environmental, social, and economic performance and achieving a win-win situation of environmental, social, and economic effects. Moreover, the findings of this paper also point out the way and direction for enterprises to promote green innovation development. By actively fulfilling environmental and social responsibilities, enterprises can win the trust of stakeholders including governments and investors, obtain key political and economic resources that are indispensable for green innovation, alleviate financing constraints, improve resource utilization, and enhance the output and quality of green technology innovation, and embark on a sustainable green development transformation path. 

As for the regulators, they should pay attention to the environmental information disclosure of enterprises, timely detect the possible "greenwashing" behavior of enterprises and punish these enterprises, to promote the ESG score to reflect the ESG performance of enterprises more truly and let the ESG performance promote the green innovation of enterprises in practice, that is, let the green attributes better promote the positive link between ESG score and green innovation of enterprises, and weaken the inhibiting effect of black attributes on the relationship between the two. 

As for institutional investors, they are important external stakeholders of enterprises, so enterprises will pay attention to the investment tendency of institutional investors to obtain more financing support. Therefore, institutional investors pay attention to ESG investment concepts, which is conducive to guiding enterprises to pay attention to ESG practices, pay attention to fulfilling environmental and social responsibilities, and enhance their green innovation drive.

Point 3: Improve the literature review with some papers.(see below)

REFERENCES

Auer, B.R., Schuhmacher, F., Do socially (ir)responsible investments pay? New evidence from international ESG data, (2016) Quarterly Review of Economics and Finance, 59, pp. 51-62

ESG and corporate financial performance: the mediating role of green innovation: UK common law versus Germany civil law, Chouaibi, S., Chouaibi, J., Rossi, M. EuroMed Journal of Business, 2022, 17(1), pp. 46–71

Exploring the moderating role of social and ethical practices in the relationship between environmental disclosure and financial performance: evidence from esg companies, Chouaibi, S., Rossi, M., Siggia, D., Chouaibi, J. Sustainability (Switzerland), 2022, 14(1), 209

Bi, K., Huang, P., Wang, X. Innovation performance and influencing factors of low-carbon technological innovation under the global value chain: A case of Chinese manufacturing industry, (2016) Technological Forecasting and Social Change, 111, pp. 275-284.

The effect of corporate social responsibility and the executive compensation on the implicit cost of equity: Evidence from French ESG data, Chouaibi, Y., Rossi, M., Zouari, G., Sustainability (Switzerland), 2021, 13(20), 11510

Response 3: Thanks to the reviewers' comments, we have supplemented the five papers listed for us by the reviewers, all of which can serve well as support for our arguments. In addition, we have additionally added some recent literature related to the topic to enrich our theoretical study.

Response to Reviewer 2 Comments

Point 1: Please do not use word abbreviations in the topic. Please write the full word "ESG".

Response 1: Your comments are really helpful. We have written the full word"ESG" in the topic, and the new title is "Do Environmental, Social, and Governance Scores Improve Green Innovation? Empirical Evidence from Chinese-Listed Companies". And we have given the full name of ESG when using it at the first time.

Point 2:The introduction is too weak in term of the latest study. Please write more clearly about the mechanism that how ESG scores improve green innovation.

Response 2: Your comments are really thoughtful. The advantages of environmental, social, and governance practices of companies favorably increase the intensity of green technology innovation, so the impact of ESG on corporate green innovation is mainly reflected in these three aspects. And we combine signaling theory, stakeholder theory, resource base theory and principal-agent theory to describe the mechanism that how ESG score improve green innovation.

Point 3: Please include some latest literature

Response 3: We have added more recent research literature to the latest manuscript. (see below)

REFERENCES

1.ESG and corporate financial performance: the mediating role of green innovation: UK common law versus Germany civil law, Chouaibi, S., Chouaibi, J., Rossi, M. EuroMed Journal of Business, 2022, 17(1), pp. 46–71

2.Adomako S, Tran M D. Environmental collaboration, responsible innovation, and firm performance: The moderating role of stakeholder pressure[J]. Business Strategy and the Environment, 2022, 31(4): 1695-1704.

3.Yuan B, Cao X. Do corporate social responsibility practices contribute to green innovation? The mediating role of green dynamic capability[J]. Technology in Society, 2022, 68: 101868.

4.Tan Y, Zhu Z. The effect of ESG rating events on corporate green innovation in China: The mediating role of financial constraints and managers' environmental awareness[J]. Technology in Society, 2022, 68: 101906.

5.Cowan K, Guzman F. How CSR reputation, sustainability signals, and country-of-origin sustainability reputation contribute to corporate brand performance: An exploratory study[J]. Journal of business research, 2020, 117: 683-693.

6.Bardos K S, Ertugrul M, Gao L S. Corporate social responsibility, product market perception, and firm value[J]. Journal of Corporate Finance, 2020, 62: 101588.

7.Zhai Y, Cai Z, Lin H, et al. Does better environmental, social, and governance induce better corporate green innovation: The mediating role of financing constraints[J]. Corporate Social Responsibility and Environmental Management, 2022, 29(5): 1513-1526.

8.Buallay, A., Hamdan, R., Barone, E., & Hamdan, A. (2022). Increasing female participation on boards: Effects on sustainability reporting. International Journal of Finance and Economics, 27(1), 111–124.

9.The effect of corporate social responsibility and the executive compensation on implicit cost of equity: Evidence from French ESG data, Chouaibi, Y., Rossi, M., Zouari, G., Sustainability (Switzerland), 2021, 13(20), 11510

10.Tolliver C, Fujii H, Keeley A R, et al. Green innovation and finance in Asia[J]. Asian Economic Policy Review, 2021, 16(1): 67-87.

11.Chouaibi S, Rossi M, Siggia D, et al. Exploring the moderating role of social and ethical practices in the relationship between environmental disclosure and financial performance: evidence from ESG companies[J]. Sustainability, 2022, 14(1): 209.

12.Tan Y, Zhu Z. The effect of ESG rating events on corporate green innovation in China: The mediating role of financial constraints and managers' environmental awareness[J]. Technology in Society, 2022, 68: 101906.

Point 4: I am really astonished why the authors do not explain the base line method? please explain the details. For example, why we are using this methodology? advantages? comparing to another methodology.

Response 4: Your comments are really useful. The reasons for using this approach are explained in our latest manuscript. In the paper, the number of N is much larger than the number of T，so our data are short panel data. And a baseline regression model can represent the significant relationship between the independent variable ESG score and the dependent variable green innovation level. The decision of fixed and random effect lies on the result of Hausman's test. This test presents a significant result which proves the use of fixed effect regression analysis(Hausman's test results are omitted in the text). So we use this model to control for year-fixed effects, industry-fixed effects, and province-fixed effects to control for the effect of unobservable factors at the industry and province levels overtime on the relationship between ESG score firms and green innovation level, and to city-level clustering. In addition, we can use the model to further examine the mechanisms and moderators of ESG scores affecting firms' green innovation. 

Point 5: The method which authors used in this paper is too much old but its ok to calculate the statics of the variables. Instead, authors can use CS-ARDL (which can calculate short and long run) Quantile and Quantile Approach, Quantile via movement.

Response 5: We appreciate it very much for this good suggestion. The CS-ARDL model allows for the determination of long and short term trends in panel data and could be a good addition to the paper. And we have been actively looking for an answer on using CS-ARDL model, while the short panel data I used does not support the use of the model because the sample in our study is firm level, the sample size is too large, and we selected many firm level control variables in the manuscript. 

Point 6: Secondly author should use second generation panel unit root, otherwise the results will be called spurious regression. Authors should use cross-sectional dependence (CD) test.

Response 6: We appreciate it very much for this good suggestion, and we have done it according to your ideas. The existence of unit roots in panel data can have serious consequences such as pseudo-regression, so we use both the Im-Pesaran-Shin test and Levin-Lin-Chu test to perform unit root tests for each variable to ensure the smoothness of each variable. As for the CD test, we have considered the soundness of the article structure and referring to some papers, we did not put this part of the analysis into the article. In general, if the data is large T small N, the article needs to consider the problem of spurious regression. However, we choose company-level data, where N is much larger than T, so the pseudo-regression problem is not very serious and can be solved by two-way fixed effects. The reviewers' comments are very good, and we will further revise it according to the comments if necessary.

Point 7:Please the discuss the results in more good way like why its negative, positive and significant? what is Economics behind is that?

Response 7：Your comments are really helpful. We have added the discussion about the results in more good way and we have added the economics behind it. We add a detailed discussion of the significance of ESG scores positively influencing green innovation for firms, governments, regulators, and investors, as well as the importance of firms playing up green attributes and increasing environmental disclosure when engaging in ESG practices. The additions are marked in red in the new manuscript.

Point 8: Please check the English. Grammar and sentence. 

Response 8：Thank you very much for finding this error. We are sorry for this grammar problem and have corrected it according to your suggestion. We have double-checked the English to polish the language throughout the manuscript. 

We tried our best to improve the manuscript and made some changes in the manuscript.

We appreciate for Editors/Reviewers' warm work earnestly, and hope that the correction will meet with approval. Once again thank you very much for your comments and suggestions.

Yours sincerely

Danni Chen

---

## [Decision Letter · Decision Letter 1]

3 May 2023

Do Environmental, Social, and Governance Scores Improve Green Innovation? Empirical Evidence from Chinese-Listed Companies

PONE-D-22-33103R1

Dear Dr. Chen,

We’re pleased to inform you that your manuscript has been judged scientifically suitable for publication and will be formally accepted for publication once it meets all outstanding technical requirements.

Kind regards,

José Antonio Clemente Almendros, PhD

Academic Editor

PLOS ONE

Additional Editor Comments (optional):

Reviewers' comments:

Reviewer's Responses to Questions

**Comments to the Author**

1. If the authors have adequately addressed your comments raised in a previous round of review and you feel that this manuscript is now acceptable for publication, you may indicate that here to bypass the “Comments to the Author” section, enter your conflict of interest statement in the “Confidential to Editor” section, and submit your "Accept" recommendation.

Reviewer #1: All comments have been addressed

Reviewer #2: All comments have been addressed

2. Is the manuscript technically sound, and do the data support the conclusions?

Reviewer #1: Yes

Reviewer #2: Yes

3. Has the statistical analysis been performed appropriately and rigorously? 

Reviewer #1: Yes

Reviewer #2: Yes

4. Have the authors made all data underlying the findings in their manuscript fully available?

Reviewer #1: Yes

Reviewer #2: Yes

5. Is the manuscript presented in an intelligible fashion and written in standard English?

Reviewer #1: Yes

Reviewer #2: Yes

6. Review Comments to the Author

Reviewer #1: I thank you for all chages. In my personale opinion, this version the paper is well structured and ready for publication.

Reviewer #2: I thoroughly read the paper. The authors revised the paper according to my comments. So accept the paper in current form.

7. PLOS authors have the option to publish the peer review history of their article (what does this mean?). If published, this will include your full peer review and any attached files.

Reviewer #1: **Yes: **Matteo Rossi

Reviewer #2: **Yes: **Faheem Ur Rehman

---

## [Editor Report · Acceptance letter]

16 May 2023

PONE-D-22-33103R1 

Do Environmental, Social, and Governance Scores Improve Green Innovation? Empirical Evidence from Chinese-Listed Companies 

Dear Dr. Chen:

I'm pleased to inform you that your manuscript has been deemed suitable for publication in PLOS ONE. Congratulations! Your manuscript is now with our production department. 

Kind regards, 

on behalf of

Dr. José Antonio Clemente Almendros 

Academic Editor

PLOS ONE